# Comparison of Head Strike Incidence under K1 Rules of Kickboxing with and without Helmet Protection—A Pilot Study

**DOI:** 10.3390/ijerph20064713

**Published:** 2023-03-07

**Authors:** Łukasz Rydzik, Wojciech Wąsacz, Tadeusz Ambroży, Tomasz Pałka, Ewa Sobiło-Rydzik, Marta Kopańska

**Affiliations:** 1Institute of Sports Sciences, University of Physical Education, 31-571 Kraków, Poland; 2Department of Physiology and Biochemistry, Faculty of Physical Education and Sport, University of Physical Education, 31-571 Kraków, Poland; 3Independent Researcher, 35-326 Rzeszów, Poland; 4Department of Pathophysiology, Institute of Medical Sciences, Medical College of Rzeszów University, 35-959 Rzeszów, Poland

**Keywords:** protective helmet, kickboxing, head injuries, brain injuries, health protection, fight analysis

## Abstract

Background: Kickboxing is a combat sport that encompasses various forms of competition. K1 kickboxing is conducted without any restrictions on the force of strikes, and the bout can end prematurely through a knockout. Headgear has been introduced in amateur kickboxing to safeguard the head. However, scientific studies have shown that despite their use, serious head injuries can still occur. The aim of this study was to evaluate the temporal structure of the bout by calculating the number of head strikes in K1 kickboxing bouts with and without headgear. Methods: Thirty K1 kickboxing bouts were analyzed, with 30 participants included in the study. The fights were conducted according to the World Association Kickboxing Organization (WAKO) rules. The bouts consisted of three rounds of 2 min each, with a 1 min break between rounds. Sparring pairs were arranged according to weight categories. The first bouts were conducted without headgear, and two weeks later, the fights were repeated with WAKO-approved headgear. The number of head strikes was assessed retrospectively by analyzing video recordings of the bouts, categorizing strikes as hand or foot strikes, and differentiating between strikes that hit the head directly or indirectly. Results: The results showed statistically significant differences between bouts with and without headgear in terms of the number of strikes to the head (*p* = 0.002), strikes directly to the head (*p* < 0.001), all hand strikes to the head (*p* = 0.001), hand strikes directly to the head (*p* = 0.003), and foot strikes directly to the head (*p* = 0.03). Higher values were observed in bouts with headgear. Conclusions: Headgear increases the probability of direct strikes to the head. Therefore, it is important to familiarize kickboxers with the use of headgear in their sport to minimize head injuries.

## 1. Introduction

Kickboxing is a combat sport that involves physical confrontation between two athletes [1,2]. It presents many variations in competitions with divisions of the range of techniques used and of the force of their execution [3,4,5]. K1 rules represent one of the modern directions in the development of martial arts, which is a hybrid of classical kickboxing with additional offensive elements of Muay Thai [4]. According to the World Kickboxing Organization (WAKO), K1 rules comprise some of the most popular forms of fighting in kickboxing [6]. K1 fights take place in a standing position and allow the use of both hand and leg strikes, including classic boxing punches, as well as the spinning back fist, jumping punch, and various types of kicks: circular, frontal, lateral, descending, hooking, and knee strikes [6]. Unconventional techniques characteristic of Muay Thai, such as the use of knee strikes, are also included. However, elbow strikes and clinching are not allowed. Fights consist of three rounds, each lasting two minutes, with a one-minute break between rounds [7]. Fights can end in one of three ways: knockout, technical knockout, or a decision by judges. In the case of a knockout, when a fighter is knocked out and unable to continue the fight, his opponent is declared the winner. In the case of a technical knockout, the fight is stopped when one of the fighters is significantly weakened and unable to continue. A judge’s decision is announced after the end of the third round and is usually based on the number of points earned by each fighter during the fight. Points are awarded for precise and effective strikes, activity in the ring, and defense against the opponent’s attacks [4,6,7]. Athletes must also adhere to the rules of fair play and respect for their opponent, and any inappropriate behavior is penalized by the judges [5]. K1 has a significant influence on the world of combat sports and commercial trends. The fighting rules are transparent and easy to understand, and the confrontation is very dynamic and full of many twists and turns, which contributes to the growing popularity of this discipline among combat sports fans and beyond [8,9].

Competing in kickboxing requires proper physical preparation, especially in terms of speed, strength, and special endurance [10,11]. In amateur competitions, fighters are protected by special head guards (helmets), which aim to minimize the risk of injury and potential trauma [12]. It is important to remember that kickboxing is a contact sport, and even with protective gear, there is a risk of serious injuries [13]. During ring competitions, athletes often win their fights by a knockout [14]. A knockout in combat sports, including kickboxing, is a situation where a fighter loses consciousness during the fight after being struck, which results in the automatic termination of the bout [15,16]. Due to a strong punch that hits the head or torso, the fighter’s brain may be damaged, leading to the temporary or even permanent loss of consciousness [17,18]. Knockout is a very dangerous situation for the fighter’s health because a loss of consciousness can lead to serious consequences, such as permanent brain damage, cognitive dysfunction, and even death [19,20]. In combat sports, including kickboxing, ensuring the safety of athletes is an utmost priority. Therefore, rules have been established as part of fight regulations aimed at minimizing the risk of injuries and ensuring swift and effective action in case of hazardous situations in terms of the athlete’s health [21]. In the event of a knockout, the referee immediately stops the fight, and the athlete is examined by a physician to determine their health status. If the athlete is unable to continue the fight, they are declared the loser [6]. A knockout can also require immediate medical assistance, and further medical care may be necessary after the end of the fight [22]. Knockouts are situations that should be avoided in combat sports from a health perspective. Therefore, athletes must be physically and mentally well-prepared and use appropriate techniques and tactics to minimize the risk of such a situation [23]. A professional K1 athlete is exposed to tremendous physical and mental pressure. Their training process is very long and demanding, consuming a lot of energy and requiring maximum concentration, with frequent injuries [22]. Athletes typically pay attention only to serious injuries, trying not to show pain or weakness. They often push themselves to train on the edge of their abilities, taking significant risks [24].

There are many scientific studies verifying changes in the brain resulting from training and boxing competitions [25,26,27,28]. Therefore, amateur kickboxing is characterized by the use of protective helmets to prevent injuries [8]. In many combat sports, such as boxing, kickboxing, or MMA, protecting the head from blows is crucial for the safety of athletes [29,30,31,32]. The helmet is one of the most critical elements of head protection, but it is not the only way to protect against blows [33]. In many cases, using a helmet can cause disorientation, improper tactics, and reduced visibility. Many researchers have addressed the issue of helmet use in combat sports [34,35], but comprehensive analyses comparing fights with and without helmets are lacking. Recent studies in kickboxing focus mainly on the analysis of physiological variables during fights [36,37,38,39] and the implementation of innovative training methods [40,41,42,43]. It is worth noting that a comparison of the number of blows to the head in fights with and without a helmet could encourage a more in-depth verification. Therefore, the aim of this study was to assess the temporal structure of a kickboxing match by calculating the number of head blows landed during a K1 kickboxing fight with and without a helmet.

## 2. Materials and Methods

### 2.1. Study Design

Thirty kickboxing fights under K1 rules were analyzed, evaluating 30 fighters divided into 15 fights with protective helmets and 15 fights without them. All fights took place on a professional 610 by 610 cm ring. The fighters followed the regulations of the World Association of Kickboxing Organizations (WAKO), consisting of 3 rounds, each lasting 2 min [7]. All the fighters wore 10 oz gloves and foot and shin guards, as well as mouth guards. The fights were recorded using a sports GOPRO HERO10 camera (GOPRO San Marino, CA, USA), which captured the entire ring. The camera was chosen after testing for its high recording quality and image stabilization function. Sparring pairs were established by the research coordinator according to weight categories (−71 kg, −75 kg, −81 kg, −86 kg, and −91 kg). The first fights were conducted without helmets, and two weeks later, the fights were repeated with helmets that met WAKO criteria (Figure 1). Based on the recordings, the number of techniques aimed at the head was evaluated, with a division of hand and foot techniques and with a similar division of techniques that directly hit the head. All participants gave written consent for the study and were thoroughly informed about the experiment. The study obtained approval from the bioethical commission. The data were entered into a statistical spreadsheet.

In addition, after each fight with the helmet, the research supervisor conducted a short direct interview with the fighters, asking about their sense of threat during the fight, as well as their comfort, visibility, and overall feeling while wearing the helmet. The responses given by the fighters were recorded and then subjected to statistical analysis.

### 2.2. Participants

The study was conducted on a group of 30 male kickboxing athletes fighting in the K1 format. The participants were aged between 20 and 35 years old. The sample size was calculated using the G*Power program. The inclusion criteria for the study were a minimum of 5 years of sports experience, active participation in competitions, a positive recommendation from the coach, and a good health status. The exclusion criteria were injuries, insufficient competition experience, a lack of recommendation from the coach, and a refusal to participate in the study. The description of the study group is presented in Table 1.

### 2.3. Statistical Analysis Methods

A statistical analysis of the collected data was performed on logarithmic data in the PQStat program. The basic descriptive statistics were calculated, including the arithmetic mean, standard deviation, minimum, maximum, and lower and upper quartile. The significance of differences was tested using the paired *t*-test. The choice of test was determined by the fulfillment of the assumption of normal distribution, which was tested using the Shapiro–Wilk test. Additionally, the effect size was calculated according to Cohen’s d, with the following values: small effect d = 0.20, moderate effect d = 0.50, strong effect d = 0.80. The level of statistical significance was set to *p* < 0.005.

## 3. Results

Statistically significant differences were found between the number of delivered kicks and punches during fighting with and without a helmet, with a moderate effect size. The results show that the participants delivered more strikes when wearing a helmet, but that this also led to significantly more direct hits to the head, which showed a statistically significant difference of *p* < 0.001 and a strong effect size. (Table 2, Figure 2).

Statistically significant differences were found between fighting without a helmet and with head protection provided by the helmet. The participants who fought wearing a helmet received a significantly higher number of strikes. The differences are visible in all the descriptive statistics presented (Table 3).

Similarly, in the case of the use of kicking techniques, the fighters hit more frequently while wearing the helmet with a statistical significance of *p* < 0.03 (Table 4).

### Interview

From the interview, it appears that all the examined athletes declared that they felt less threatened by their opponent when fighting with a helmet. However, 100% of the respondents also reported that fighting with a helmet caused discomfort and limited visibility (Figure 3).

## 4. Discussion

The aim of this study was to determine the number of techniques directly targeting the head and to compare the number of techniques received when using head protection (a helmet) versus when using no protection. In this context, it is worth noting that combat sports such as kickboxing require certain technical and tactical skills from athletes in order to fight effectively and minimize the risk of injury. However, depending on the rules of the fight and the equipment used by the athletes, the risk of injury and trauma can vary significantly.

The statistical analysis showed that fighting with a helmet resulted in greater engagement, manifested by a higher number of techniques being executed. During the fight without a helmet, athletes statistically hit less often than during the fight with a helmet. In our research, we obtained the subjective opinions of the tested athletes through direct interviews. The participants reported feeling less threatened by their opponent when fighting with a helmet, which led them to focus more on offensive actions, neglecting certain defensive aspects. The opposite proportion was noted in the opinions regarding confrontations without helmets, where the participants were clearly more focused and concentrated on effective defense.

On one hand, using a helmet can increase the sense of safety for athletes, which in turn leads to an increase in the number of techniques being executed. On the other hand, a greater number of blows and techniques that target the head area can increase the risk of injury and trauma [9]. It is worth noting that not only does the number of blows and techniques affect the risk of injury in kickboxing, but so do the force and precision of the strikes [5]. Research has shown that head injuries are the most common injuries in combat sports, and that brain injuries can lead to serious health problems such as memory disorders, balance problems, or even permanent brain damage [44,45]. Protecting the head with a helmet may potentially enlarge the target for the opponent, as the circumference of the head increases due to the structure of the helmet, making it an easier target to hit [45]. Additionally, athletes may not be properly prepared to fight with a helmet, which can lead to tactical errors [46].

In our own research, through direct interviews, each participant was asked an open-ended question about their personal opinion on fighting with and without a helmet. All participants stated that they preferred to fight without protective headgear, citing better visibility, which they believed was related to a better reaction time and better timing in technical (offensive-defensive) actions. They also reported feeling more comfortable in aspects such as movement, breathing, and experiencing the effects of thermoregulation on the head, which contributed to a better overall sense of well-being during the fight.

Kickboxing fights based on K1 rules are characterized by rules that do not limit the force of the techniques used. As a result, they carry the risk of serious injuries, especially to the brain [47]. Scientific studies conducted by Stormezand et al. indicate that changes in the brain among kickboxing athletes may be a consequence of numerous repeated blows to the head. Therefore, it can be concluded that fighting without a helmet, which results in a greater number of received blows, may have a more negative impact on head injuries. Similar conclusions were drawn from research conducted on boxers, where amateur boxers experienced frequent injuries (mainly cuts, bruises, and sprains), as well as further consequences after the injury, such as problems with coordination or memory [48]. A higher number of punches and kicks aimed at the head may indicate a higher likelihood of injury or a very serious injury [25,26]. This is also significantly correlated with the possibility of losing the fight by a knockout [49]. The helmet during kickboxing fights aims to protect the head from injuries, particularly the skin and skull bones. However, it is not designed to provide the full protection of the brain from injuries. Brain injuries are a serious problem in combat sports such as boxing and kickboxing, as blows to the head can cause a concussion [13,50,51]. A study conducted by Gessel et al. showed that kickboxers who had head injuries, regardless of whether they wore a helmet, experienced a similar level of headaches, dizziness, balance problems, and memory difficulties. This study suggests that helmets do not provide full protection against brain injuries in kickboxing [52]. Another study conducted by McCrory et al. showed that using helmets in boxing and other combat sports reduces the number of cuts and bruises, but has no effect on reducing the number of brain injuries [53]. Therefore, in the present study, statistically significant differences were shown between the percentage ratios of the techniques introduced and the ones that hit the target. The results show that about 47% of the techniques executed during the fight with a helmet involved hits directly to the head, while during the competition without a helmet, the number decreased to 27%. It is therefore intriguing if, despite its protective functions, the use of a helmet does not cause greater injuries inside the head than not using one. A greater number of hits may be caused by the limited visibility when wearing a helmet. Additionally, athletes wearing a helmet may feel safer and pay less attention to protecting their head. This study serves as a pilot for a more detailed verification of the problem through serious medical analysis.

### Limitations

The main limitation of this study is that there was an inability to check for head vibrations after a blow was received. It would also be valuable to check brain function by conducting professional medical tests.

## 5. Conclusions

Athletes fighting in helmets are exposed to a greater number of techniques directly hitting the head, which may have an impact on the occurrence of injuries or serious harm. Additionally, further research is required to verify the detailed changes in the brain that may occur as a result of K1 kickboxing competitions using professional medical analysis.

### Practical Implications

During the specialized training process of kickboxing athletes preparing for K1 competitions, trainers should familiarize athletes with helmets and adjust them to the appropriate tactics for competing with full protection.

## Figures and Tables

**Figure 1 ijerph-20-04713-f001:**
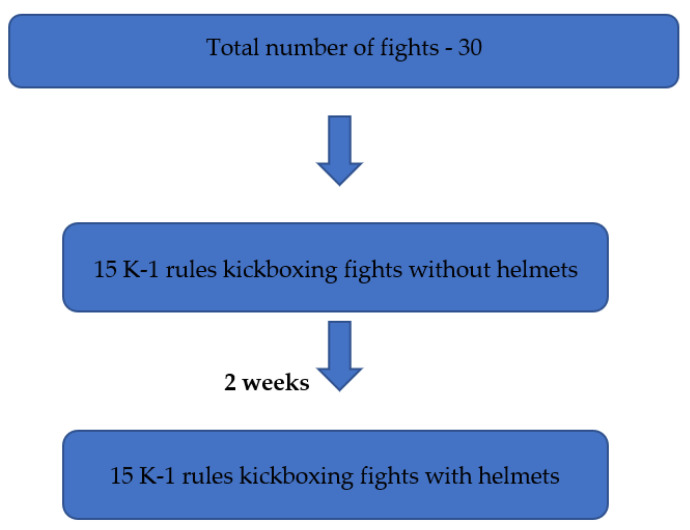
Study design.

**Figure 2 ijerph-20-04713-f002:**
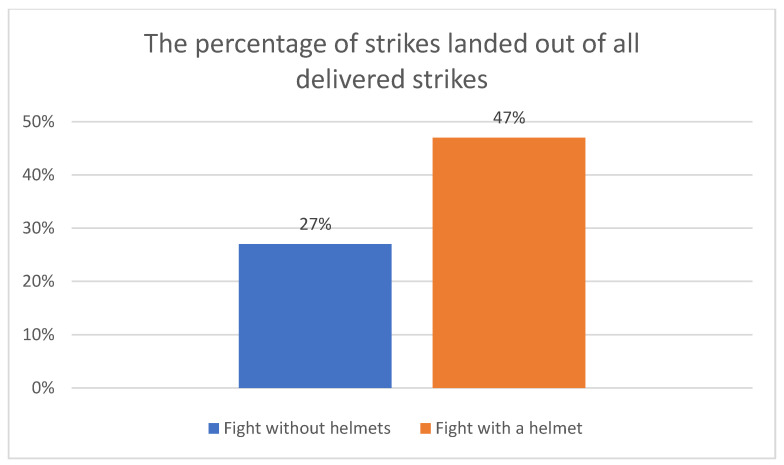
The percentage of strikes landed out of all delivered strikes.

**Figure 3 ijerph-20-04713-f003:**
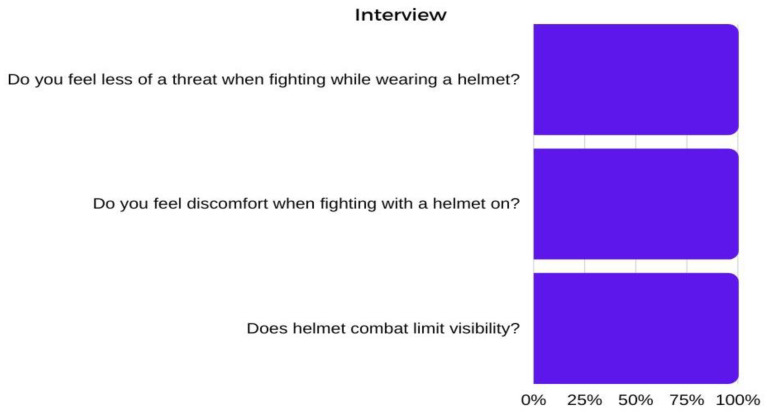
Results of the interview.

**Table 1 ijerph-20-04713-t001:** Description of the studied group.

	X	SD	Min	Max	Q1	Q3
Age (years)	26.4	4.85	20	35	23	29.75
Body Mass (kg)	80.7	3.97	76	90	78.5	82.25
Body Height (cm)	182	3.88	175	186	179	186

n—number of subjects, X—mean, SD—standard deviation, Min—minimum, Max—Maximum, Q1—lower quartile, and Q3—upper quartile.

**Table 2 ijerph-20-04713-t002:** Descriptive and comparative statistics of fights with and without a helmet in relation to hand and foot techniques.

	Fight without Helmets						Fight with a Helmet								
	X	SD	Min	Max	Q1	Q3	X	SD	Min	Max	Q1	Q3	t	*p*	d
The total number of strikes delivered	96	38.59	39	174	72	111	113.2	30.64	62	179	97	131	**−4.14**	**0.002**	**0.50**
The number of strikes landed	24.8	7.74	13	39	17	29	52.7	17.38	21	93	40.75	60.25	**−5.33**	**<0.001**	**2.22**
The percentage of strikes landed out of all delivered strikes.	0.27	0.06	0.19	0.38	0.22	0.33	0.47	0.13	0.26	0.69	0.39	0.57	**−7.38**	**<0.001**	**211**

*p*-value—level of statistical significance, t-value—result of the *t*-test, d-value—effect size. Bold values indicate statistically significant results.

**Table 3 ijerph-20-04713-t003:** Descriptive and comparative statistics of fights with and without a helmet in relation to hand techniques.

	Fight without Helmets						Fight with a Helmet								
	X	SD	Min	Max	Q1	Q3	X	SD	Min	Max	Q1	Q3	t	*p*	d
The number of hand techniques delivered	88.8	35.92	38	155	62	104	106.6	30.43	58	170	86	124	**−4.57**	**0.001**	**0.54**
The number of hand techniques landed	24	7.34	12	36	17	27	45.6	18.08	21	85	33	55	**−4.01**	**0.003**	**1.70**

Bold values indicate statistically significant results.

**Table 4 ijerph-20-04713-t004:** Descriptive and comparative statistics of fights with and without a helmet in relation to kicking techniques.

	Fight without Helmets						Fight with a Helmet								
	X	SD	Min	Max	Q1	Q3	X	SD	Min	Max	Q1	Q3	t	*p*	d
The number of kicking techniques delivered	6.6	4.70	1	18	3	9	6.6	2.84	1	11	4	9	0	1	0.00
The number of kicking techniques landed	0.8	0.99	0	3	0	1	1.56	1.43	0	5	1	2	**−2.29**	**0.03**	**0.63**

Bold values indicate statistically significant results.

## Data Availability

All data was included in the manuscript.

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
