# Peer review of "Comparison of Head Strike Incidence under K1 Rules of Kickboxing with and without Helmet Protection—A Pilot Study"

_ijerph, 2023, doi:10.3390/ijerph20064713_

Round 1

Reviewer 1 Report

This article contributes to a spreading concern in combat, i.e., minimizing the risk of injury and potential trauma (notably in Kickboxing). It is well documented and could lead to a better understanding of head guard use and a proactive approach to mitigating repetitive knockout's possible short and long-term effects. Such biomechanical with head kinematics studies are needed to quantify the magnitude and frequency of HAE sustained in training and game exposures.

But I have identified a few items I would recommend to better target the studies' boundaries and translate them into material clarification and potential recommendations or future development.

A)     Line 110-1120: Could you give more information on the "specialized GOPRO HERO10 camera"? It is also central to the scientific method behind the study and how the number of techniques aimed at the head was evaluated and determined (from the recording)—simple observations or else.

B)     Line 140-145: Concerning the responses given by the fighters were recorded and then subjected to statistical analysis. Were they multiple choices questions or open questions? And if it was open questions. Please clarify how they were "converted" into statistical data.

C)     Pages 5-7: Considering the size of the figures and tables, there is no need to change the pages' orientation. A regular "Portrait" format can be kept (rather than the Landscape format used for these three pages). That would add to the clarity of the document.

D)     In the Discussion section (4.), Line 44-48: Please consider simplifying the sentence, dividing it like this (or else): "In our research, we diagnosed the subjective opinion of the tested athletes through direct interviews. Participants reported feeling less threatened by their opponent when fighting with a helmet, which led them to focus more on offensive actions, neglecting certain defensive aspects." It's an important observation from which a brief version  might be carried into the conclusion.

E)     Page 9, lines 93-97: Please elaborate on these interesting observations. What could explain the “significant differences”? Is it just “in the present” study as written, while  47% vs. 27% is effectively significant, but what would be the rationale from the authors' standpoint if it is beyond a single study? The reader would benefit from these insights in the discussions. Therefore, in the present study, statistically significant differences were shown between the percentage ratio of techniques introduced and the ones that hit the target. By the way, this statement: “The results show that about 47% of the techniques executed during the fight with a helmet hit directly to the head, while during the competition without a helmet, the number decreases to 27%” could also be referred to as a conclusion if further elaborate.

F)      Page 9, lines 97-100. This sentence deserves more clarification “It is therefore intriguing whether, despite its protective functions, the use of a helmet does not cause greater injuries inside the head than without its application. This study serves as a pilot for more detailed verification of the problem through serious medical analysis.”  What is the potential justification behind it? Was it because of the more considerable number of direct hits (to the head with a Helmet), the injuries should also be more significant, despite the protection, or is it something else?

G)    Also, a language revision (while not crucial) might be fruitful.  Here are a few examples of typos and format adjustments:

Line 109: World Association of Kickboxing Organizations (WAKO)

Page 6, Line 18: The percentage (and not percent-age)

Page 8, Line 44: Please remove "own" from the sentence. "In our research" is sufficient.

Many spacing between paragraph issues in the document (one line space, two spaces, or even zero); please revisit the spacing for uniformity.

Finally, the conclusions would improve the "material" impact of the last portion of the main findings and observations  (see examples given before), from the authors' perspectives, which could be synthesized. It looks like the last sentence falls short in not opening well the following steps as seen by the authors (e.g., checking brain function by conducting professional medical tests). It's an important topic, more studies are needed and the readers would surely welcome some orientation or recommendations.

Author Response

Dear Reviewer,

Thank you very much for your time and valuable comments, which all have been considered and incorporated. The detailed list of responses is given below. We hope that the modifications and explanation will be acceptable for you.

Yours sincerely,

Rydzik, corresponding author

This article contributes to a spreading concern in combat, i.e., minimizing the risk of injury and potential trauma (notably in Kickboxing). It is well documented and could lead to a better understanding of head guard use and a proactive approach to mitigating repetitive knockout's possible short and long-term effects. Such biomechanical with head kinematics studies are needed to quantify the magnitude and frequency of HAE sustained in training and game exposures.

But I have identified a few items I would recommend to better target the studies' boundaries and translate them into material clarification and potential recommendations or future development.

  1. A)Line 110-1120: Could you give more information on the "specialized GOPRO HERO10 camera"? It is also central to the scientific method behind the study and how the number of techniques aimed at the head was evaluated and determined (from the recording)—simple observations or else.

A: More information was added. There were direct observations and a detailed analysis of the recorded material, which could be stopped and slowed down. It is worth mentioning that the observations were made by competent coaches and kickboixngu judges.

  1. B)Line 140-145: Concerning the responses given by the fighters were recorded and then subjected to statistical analysis. Were they multiple choices questions or open questions? And if it was open questions. Please clarify how they were "converted" into statistical data.

A: We've added more information in the results section. Closed-ended yes or no questions were used.

  1. C)Pages 5-7: Considering the size of the figures and tables, there is no need to change the pages' orientation. A regular "Portrait" format can be kept (rather than the Landscape format used for these three pages). That would add to the clarity of the document.

A: Dear reviewer thank you for your attention. This representation is for the benefit of the review process , as the file is a PDF. If the article is accepted, this will be corrected. The final shape of the article is decided by the editorial board. 

  1. D)In the Discussion section (4.), Line 44-48: Please consider simplifying the sentence, dividing it like this (or else): "In our research, we diagnosed the subjective opinion of the tested athletes through direct interviews. Participants reported feeling less threatened by their opponent when fighting with a helmet, which led them to focus more on offensive actions, neglecting certain defensive aspects." It's an important observation from which a brief version  might be carried into the conclusion.

A: This has been corrected

  1. E)Page 9, lines 93-97: Please elaborate on these interesting observations. What could explain the “significant differences”? Is it just “in the present” study as written, while  47% vs. 27% is effectively significant, but what would be the rationale from the authors' standpoint if it is beyond a single study? The reader would benefit from these insights in the discussions. Therefore, in the present study, statistically significant differences were shown between the percentage ratio of techniques introduced and the ones that hit the target. By the way, this statement: “The results show that about 47% of the techniques executed during the fight with a helmet hit directly to the head, while during the competition without a helmet, the number decreases to 27%” could also be referred to as a conclusion if further elaborate.

A: We have added more information

  1. F)Page 9, lines 97-100. This sentence deserves more clarification “It is therefore intriguing whether, despite its protective functions, the use of a helmet does not cause greater injuries inside the head than without its application. This study serves as a pilot for more detailed verification of the problem through serious medical analysis.”  What is the potential justification behind it? Was it because of the more considerable number of direct hits (to the head with a Helmet), the injuries should also be more significant, despite the protection, or is it something else?

A: Yes, we think that with more blows received during a helmeted fight, the internal damage to the head may be greater. This is the subject of our further research, where we are conducting verification with the QEEG apparatus

  1. G)Also, a language revision (while not crucial) might be fruitful.  Here are a few examples of typos and format adjustments:

Line 109: World Association of Kickboxing Organizations (WAKO)

A: This has been corrected

Page 6, Line 18: The percentage (and not percent-age)

A: This has been corrected

Page 8, Line 44: Please remove "own" from the sentence. "In our research" is sufficient.

A: This has been corrected

Many spacing between paragraph issues in the document (one line space, two spaces, or even zero); please revisit the spacing for uniformity.

A: This has been corrected

Finally, the conclusions would improve the "material" impact of the last portion of the main findings and observations  (see examples given before), from the authors' perspectives, which could be synthesized. It looks like the last sentence falls short in not opening well the following steps as seen by the authors (e.g., checking brain function by conducting professional medical tests). It's an important topic, more studies are needed and the readers would surely welcome some orientation or recommendations.

A: This has been corrected

Reviewer 2 Report

Dear Author, the paper is well written, i have only one appointment.

Both in methods section section you report that the research supervisor conducted a short direct interview with the fighters; and in the discussion you discuss this interview, but no mention there is in the result section.

Please add also in the result section a paragraph about this interview, or delete it from this paper.

Searching in pubmed with two keywords: Head and Kikboxing you can found only 23 papers, mainly on damage due to head trauma and brain concussion.

Moreover the statement of the author that discover that use of helmet protection worse risk of have head trauma it really new, inaspected and important to all the kikboxing fighter.   The only criticism express by this reviewer is about interview that appear in the methods and discussion paragraph but not in the result section (this point even alone ia major revision if the paper was not well conduct). This missing have to be filled by the author before acception.

Author Response

Dear Reviewer,

Thank you very much for your time and valuable comments, which all have been considered and incorporated. The detailed list of responses is given below. We hope that the modifications and explanation will be acceptable for you.

Yours sincerely,

Rydzik, corresponding author

Dear Author, the paper is well written, i have only one appointment.

Both in methods section section you report that the research supervisor conducted a short direct interview with the fighters; and in the discussion you discuss this interview, but no mention there is in the result section.

Please add also in the result section a paragraph about this interview, or delete it from this paper.

A: Thank you for your review. We have highlighted the interview responses with a subsection and added a figure.

Searching in pubmed with two keywords: Head and Kikboxing you can found only 23 papers, mainly on damage due to head trauma and brain concussion.

A: Thank you for your comment, we have identified the publication you mentioned. We seem to have used all relevant studies and our research fills in the gaps.

Moreover the statement of the author that discover that use of helmet protection worse risk of have head trauma it really new, inaspected and important to all the kikboxing fighter.   The only criticism express by this reviewer is about interview that appear in the methods and discussion paragraph but not in the result section (this point even alone ia major revision if the paper was not well conduct). This missing have to be filled by the author before acception.

A: Thank you, we hope that the changes will be acceptable to you

Reviewer 3 Report

This study investigated the head strike incidence with and without helmet, which is very interesting. However, the sample number of this study is a bit small. Below are my comments and concerns:

1.       In the method part, why wait for 2 weeks to conduct the kickboxing fights with helmets? By what criteria or reference?

2.       What is the unit of each parameter in table 1?

3.       In Table 2 and 3, please use dot instead of comma.

4.       Does the field recommend a helmet for kickboxing? Please give a discussion

Author Response

Dear Reviewer,

Thank you very much for your time and valuable comments, which all have been considered and incorporated. The detailed list of responses is given below. We hope that the modifications and explanation will be acceptable for you.

Yours sincerely,

Rydzik, corresponding author

This study investigated the head strike incidence with and without helmet, which is very interesting. However, the sample number of this study is a bit small. Below are my comments and concerns:

  1. In the method part, why wait for 2 weeks to conduct the kickboxing fights with helmets? By what criteria or reference?

A: We used a two-week break between the fights to avoid the effect of the opponent's learning. By conducting the fights on the same day or the day after, they would have gone differently. Therefore, this is the procedure we adopted

  1. What is the unit of each parameter in table 1?

A: This has been corrected

  1. In Table 2 and 3, please use dot instead of comma.

A: This has been corrected

  1. Does the field recommend a helmet for kickboxing? Please give a discussion

A: We are conducting further research in this direction through QEEG verification. Current research shows that the use of a helmet does not increase the protection of the head but results in an easier hit and the concussion that goes with it. Already from this research, it is apparent that fighters receive some blows to the head when they fight wearing a helmet. This may be due to the limitations caused by wearing a helmet or to a reduced attention to defence.

Round 2

Reviewer 3 Report

The authors have addressed my questions.